# Singularity Avoidance Control of a Non-Holonomic Mobile Manipulator for Intuitive Hand Guidance [†]

**Matthias Weyrer [1],\*, Mathias Brandstötter [1] and Manfred Husty [2]**

[1] Joanneum Research, Institute for Robotics and Mechatronics, Lakeside B08a, 9020 Klagenfurt am Wörthersee, Austria; mathias.brandstoetter@joanneum.at

[2] Unit Geometry and CAD, University of Innsbruck, 6020 Innsbruck, Austria; manfred.husty@uibk.ac.at

\* Correspondence: matthias.weyrer@joanneum.at; Tel.: +43-316-876-2010

[†] This paper is an extended version of our paper published in Weyrer, M.; Brandstötter, M.; Mirkovic, D. Intuitive Hand Guidance of a Force-Controlled Sensitive Mobile Manipulator. In Proceedings of the IFToMM Symposium on Mechanism Design for Robotics, Udine, Italy, 11–13 August 2018.

**Abstract:** Mobile manipulators are robot systems capable of combining logistics and manipulation tasks. They thus fulfill an important prerequisite for the integration into flexible manufacturing systems. Another essential feature required for modern production facilities is a user-friendly and intuitive human-machine interaction. In this work the goal of code-less programming is addressed and an intuitive and safe approach to physically interact with such robot systems is derived. We present a natural approach for hand guiding a sensitive mobile manipulator in task space using a force torque sensor that is mount close to the end effector. The proposed control structure is capable of handling the kinematic redundancies of the system and avoid singular arm configurations by means of haptic feedback to the user. A detailed analysis of all possible singularities of the UR robot family is given and the functionality of the controller design is shown with laboratory experiments on our mobile manipulator.

**Keywords:** robot kinematics; robot singularity; singularity analysis; robot control; mobile manipulation; human-robot-interaction; learning by demonstration; compliance control

## 1. Introduction

The demand for highly flexible and adaptable robotic systems naturally arises within the manufacturing processes of products with high variability and small lot-sizes. This challenges also include frequently reprogramming of the robot. Traditionally, interactions between humans and robots within a shared workplace can be categorized into two distinct scenarios: a service scenario and a process scenario. In the former case, a robot is programmed and prepared for a new production process rather infrequently by highly skilled experts. In the latter case, less complicated interactions are part of the everyday work flow. This means that robot reprogramming has to be performed much more frequently by human workers with extensive domain knowledge but usually limited programming skills.

To integrate this reprogramming fluently into the workflow it must be fast and easy to use. Thus the interaction interface between human and robot is of significant importance. One well known technique is Programming by Demonstration (PbD). There are several forms of this method: (a) the positions of the work-piece itself or a special teaching object is tracked and used to plan the trajectory of the robot [1], (b) the robot is guided into the desired positions via remote control [2,3] and (c) kinestetic programming by demonstration, where the robot is compliant and can be hand-guided into the desired configurations [4,5]. In the user-centric work of [6], the trajectories that are taught to the robot system by untrained end users can be adapted in a subsequent step via a graphical user interface to obtain the desired task.

The latter mentioned teaching technique requires the robot to be compliant. There are several different sensitive robots, also known as collaborative robots or cobots that are able to perceive the interaction forces with the environment by utilizing additional sensing like joint torque sensors and control theory. With the knowledge of external forces that act on the robot, a compliant behavior can be realized which enables the ability to hand-guide the robot and allows a closer cooperation without external safety barriers [7].

While there are many publications describing compliance control for serial manipulators, e.g., [8–13] only little investigations for a whole-body compliance control of a mobile manipulators have been done. Leboutet et al. [14] proposed a technique with hierarchical force propagation for a mobile manipulator that consists of an omni-directional base and two Universal Robots UR10 serial robots. The robotic arms are covered with their special multi-modal sensor skin which allows measuring the applied external forces on the robot at several contact points. External forces whose reactive motions are inconvenient to be performed by the serial manipulator are directly projected to the mobile base. To decide which motions should be performed by the base, the manipulability ellipsoid is used. Navarro et al. [15] presented a solution for an omnidirectional base where the distribution of motion is done with optimization. They proposed a cost function that includes a measure for the manipulability, a self-defined value for the closeness-to-singularity and some additional distance and angle constraints.

Han et al. [16] point out the complexity of controlling a robot in task-space while taking singularities and joint limits into account. They present a hierarchically structured controller that uses a continuous task transition algorithm to guarantee execution of the main task while additional tasks, e.g., for singularity-avoidance, can be activated or deactivated.

In our previous work [17], we presented a control design for a whole body compliance control of the mobile manipulator but singularity avoidance was not taken into account. Since we control the velocities of the end effector (EE) in task space, singular configurations are problematic. In a singular configuration the inverse kinematic on velocity level cannot be solved at all or results in infinity joint velocities. Also approaching a configuration close to a singularity may result in very high joint speeds, which could be dangerous for humans close to the robot, and must be avoided. We extended our previous work by analyzing all possible singularities of the Universal Robots family with focus on the model UR10, which is used on our mobile manipulator CHIMERA. We also included a singularity-avoidance strategy in our control structure by applying haptic feedback to the user before approaching singular configurations and present the results of conducted laboratory experiments.

This paper is organized as follows: The kinematics and especially the singularity analysis of the serial manipulator UR10 is given in Section 2, the control structure is discussed in Section 3, including the motion-distribution between mobile base and serial manipulator and our proposed strategy to avoid approaching singular arm configurations. Experimental results are shown in Section 4 and a conclusion and outlook for future work is given in Section 5.

## 2. Kinematics and Singularity Analysis

A mobile manipulator is an effective tool to accomplish tasks, e.g. the manipulation of objects in space. It is a combination of a serial manipulator and a mobile robot, which greatly expands the manipulator's workspace and thus increases the system's performance. For analysis purposes, such systems can often be split into two components, a mobile platform and a manipulator arm. The studies in this paper focus on a mobile manipulator called CHIMERA, which consists of a MiR platform (differential drive) and a UR10 (6 DoF) serial arm.

### 2.1. Kinematics

Mobile wheeled platforms have been the subject of many studies in the past. For the kinematic description of mobile robots we refer to [18]. The kinematic relationships of the UR10 were also sufficiently investigated [19], although it is pointed out that the kinematic chain has an offset wrist.

### 2.2. Singularity Analysis of the UR Robot

For the computation of all singularities of the UR10 we will use the well known fact that the columns of the $6 \times 6$ Jacobian matrix $\mathbf{J}$ are the Plücker coordinates of the instantaneous locations of the rotation axes of the manipulator [20]. Using this fact one can obtain $\mathbf{J}$ without differentiation. A couple of prerequisites are noted before. We assume that the rotation axes are always the $z$-axes of the local coordinate systems. In this local coordinate system the Plücker coordinates of the revolute axes are $\mathbf{p}_i = [0, 0, 1, 0, 0, 0]^T$. To compute their coordinates in the base system the forward transformation matrices are needed. It has to be noted that the manipulator is in a singular pose when the six Plücker coordinates are linearly dependent.

Using the usual Denavit-Hartenberg (D-H) convention to describe the geometric structure of the serial manipulator [21], the forward transformation can be written as

$$\mathbf{T} = \prod_{i=1}^{6} \mathbf{M}_i \cdot \mathbf{G}_i \tag{1}$$

where

$$\mathbf{M_i} = \begin{bmatrix} 1 & 0 & 0 & 0 \\ 0 & \cos q_i & -\sin q_i & 0 \\ 0 & \sin q_i & \cos q_i & 0 \\ 0 & 0 & 0 & 1 \end{bmatrix}, \quad \mathbf{G_i} = \begin{bmatrix} 1 & 0 & 0 & 0 \\ a_i & 1 & 0 & 0 \\ 0 & 0 & \cos \alpha_i & -\sin \alpha_i \\ d_i & 0 & \sin \alpha_i & \cos \alpha_i \end{bmatrix}.$$

The joint positions of the serial manipulator are given by $q_i$ as depicted in Figure 2 and the constant D-H parameters are given by $a_i$, $d_i$ and $\alpha_i$. To transform the Plücker coordinates the line transform matrix $\overline{\mathbf{T}}$ is needed. When the forward transformation matrix is written as

$$\mathbf{T} = \begin{bmatrix} 1 & 0 \\ \mathbf{a} & \mathbf{A} \end{bmatrix}, \qquad \mathbf{a} \dots 3 \times 1 \text{ translation vector}, \ \mathbf{A} \dots 3 \times 3 \text{ rotation matrix}$$

then the line transform matrix is

$$\overline{\mathbf{T}} = \begin{bmatrix} \mathbf{A} & \mathbf{0} \\ \mathbf{a}^{\times} \mathbf{A} & \mathbf{A} \end{bmatrix}. \qquad \mathbf{a}^{\times} \dots \text{skew symmetric matrix belonging to translation vector } \mathbf{a}$$

To compute the Plücker coordinates of a specific rotation axis only those parts of the forward kinematics will be needed which transform up the axis whose location has to be found. We denote the partial transformations by

$$\mathbf{T}_j = \prod_{i=1}^{j} \mathbf{M}_i \cdot \mathbf{G}_i, \qquad j = 1, \dots, 5$$

and by $\mathbf{y}_1 = [0, 0, 1, 0, 0, 0]^T$ the Plücker coordinates of the first rotation axis. Then the remaining five Plücker coordinates are obtained by

$$\mathbf{y}_k = \mathbf{T}_{k-1} \cdot \mathbf{y}_1. \qquad k = 2, \dots, 6 \tag{2}$$

The six Plücker coordinates can now be assembled to the $6 \times 6$ Jacobian matrix $\mathbf{J}$:

$$\mathbf{J} = [\mathbf{y}_1, \mathbf{y}_2, \mathbf{y}_3, \mathbf{y}_4, \mathbf{y}_5, \mathbf{y}_6] \tag{3}$$

A necessary and sufficient condition for the manipulator being in a singularity is: $\det \mathbf{J} = 0$. Due to the simplicity of the design of the manipulator this determinant can be computed without assigning all D-H parameters. The resulting equation becomes very well laid out when all angles in the forward transformation are written in algebraic values. This is achieved by performing half tangent substitution: $\cos q_i = \frac{1 - v_i^2}{1 + v_i^2}$, $\sin q_i = \frac{2v_i}{1 + v_i^2}$, $\cos \alpha_i = \frac{1 - al_i^2}{1 + al_i^2}$, $\sin \alpha_i = \frac{2al_i}{1 + al_i^2}$. The essential D-H

parameters that determine the UR family of robots are $a_1 = 0$, $d_2 = 0$, $d_3 = 0$, $a_4 = 0$, $a_5 = 0$, $a_6 = 0$, $al_1 = 1$, $al_2 = 0$, $al_3 = 0$, $al_4 = -1$, $al_5 = 1$, $al_6 = 0$. The remaining D-H parameters are not assigned and determine the type of UR robot. Computing the determinant of **J** yields

$$\det \mathbf{J} = v_3 v_5 \left[ (v_4^2 + 1)(v_3^2 + 1)(v_2 - 1)(v_2 + 1)a_2 - (v_4^2 + 1)(v_2 v_3 + v_2 + v_3 - 1)(v_2 v_3 - v_2 - v_3 - 1)a_3 \right.$$
$$\left. -(2(v_2 v_3 + v_2 v_4 + v_3 v_4 - 1))(v_2 v_3 v_4 - v_2 - v_3 - v_4)d_5 \right] = 0. \tag{4}$$

The analysis of Equation (4) reveals that $\det \mathbf{J}$ factors into three parts: $v_3 = 0$ determines the elbow singularities because then the arm is stretched out, $v_5 = 0$ yields the wrist singularities because then the fourth and the sixth axis are coplanar. The third expression belongs to the shoulder singularity and contains only the joint parameters $v_2, v_3, v_4$. When two of the three joint parameters are set, then the third can be computed via the remaining quadratic equation. When the manipulator is brought to the resulting pose then one can observe that the intersection point $P_{56}$ of the fifth and the sixth axis is on a cylinder which has the equation $x^2 + y^2 - d_4^2 = 0$ in the base coordinate system. This cylinder has a geometrical easy explanation: lets assume for a moment $v_1 = 0$, then it is obvious that the intersection point of fifth and sixth axis can only move in the plane $y = -d_4$ of the base coordinate system. This plane intersects the plane $x = 0$ which is the span of the first and the second axis in a line parallel to the z-axis in a distance $d_4$ from this axis. When the rotation about the first axis is added then this line describes the cylinder. That $P_{56}$ is located on this line in case of a shoulder singularity can be computed immediately by setting $v_1 = 0$ and solving the third polynomial of Equation (4) for, e.g., $v_4 = f(v_2, v_3)$. As the equation is quadratic in $v_4$ one obtains for arbitrary values of $v_3$ and $v_4$ two values for $v_4 = v_{41}$, $v_4 = v_{42}$. Direct computation of the location of $P_{56}$ when either $v_{41}$ or $v_{42}$ are substituted into the forward kinematic equation yields $P_{56} = [1, 0, -d_4, \pm g(v_2, v_3)]^T$. This shows that $P_{56}$ is on the intersection line of planes $x = 0$ and $y = -d_4$. Its z coordinate is determined by $g(v_2, v_3)$ which is a relatively complicated function. It gives the values of the intersection point of the circle which is the path of $P_{56}$ during the rotation about the fourth axis with the plane $x = 0$.

The forgoing description is valid for all manipulators of the UR family. When a special type is chosen, e.g. UR10, then the remaining D-H parameters are set $a_2 = 0.6127$, $a_3 = 0.5716$, $d_1 = 0.118$, $d_4 = 0.163941$, $d_5 = 0.1157$, $d_6 = 0.0922$ and the singularity equation becomes:

$$\det \mathbf{J} = v_3 v_5 \left[ 0.6127(v_4^2 + 1)(v_3^2 + 1)(v_2 - 1)(v_2 + 1) - \right]$$
$$0.5716(v_4^2 + 1)(v_2 v_3 + v_2 + v_3 - 1)(v_2 v_3 - v_2 - v_3 - 1)$$
$$-0.2314(v_2 v_3 + v_2 v_4 + v_3 v_4 - 1)(v_2 v_3 v_4 - v_2 - v_3 - v_4) \right] = 0 \tag{5}$$

The singularity surface represented by Equation (5) is shown in Figure 1.

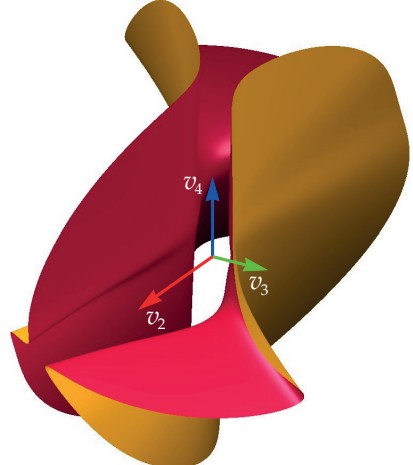

**Figure 1.** Singularity surface of shoulder singularities in the transformed joint space.

## 3. Control Strategy

The goal of the control structure is to translate the input forces and torques of the user into robot motion. We divide between the two subsystems, the mobile base and the serial manipulator on top of it. Since the combined system shows kinematic redundancies concerning the 3D task-space, the motion distribution is a main part of the proposed control structure. Additionally, virtual springs are used to generate haptic feedback to the user when pushing or pulling the mobile base. Haptic feedback is also used to avoid singular arm configurations.

We consider the serial manipulator as an open kinematic chain with $\mathbf{q}_{\text{ur}} = \begin{bmatrix} q_1 & q_2 & \dots & q_6 \end{bmatrix}^T \in \mathbb{R}^{6\times1}$ joints on top of the mobile base equipped with a differential drive, denoted as $\mathbf{q}_{\text{mir}} = \begin{bmatrix} x & y & \theta \end{bmatrix}^T \in \mathbb{R}^{3\times1}$ shown in Figure 2. All freedoms of the system are collected in $\mathbf{q}_{\text{sys}} = \begin{bmatrix} \mathbf{q}_{\text{ur}}^T & \mathbf{q}_{\text{mir}}^T \end{bmatrix}^T \in \mathbb{R}^{9\times1}$. Moreover, the redundant robot system is considered as a unit that is composed of two tightly coupled subsystems, where the coupling is established by our proposed control structure.

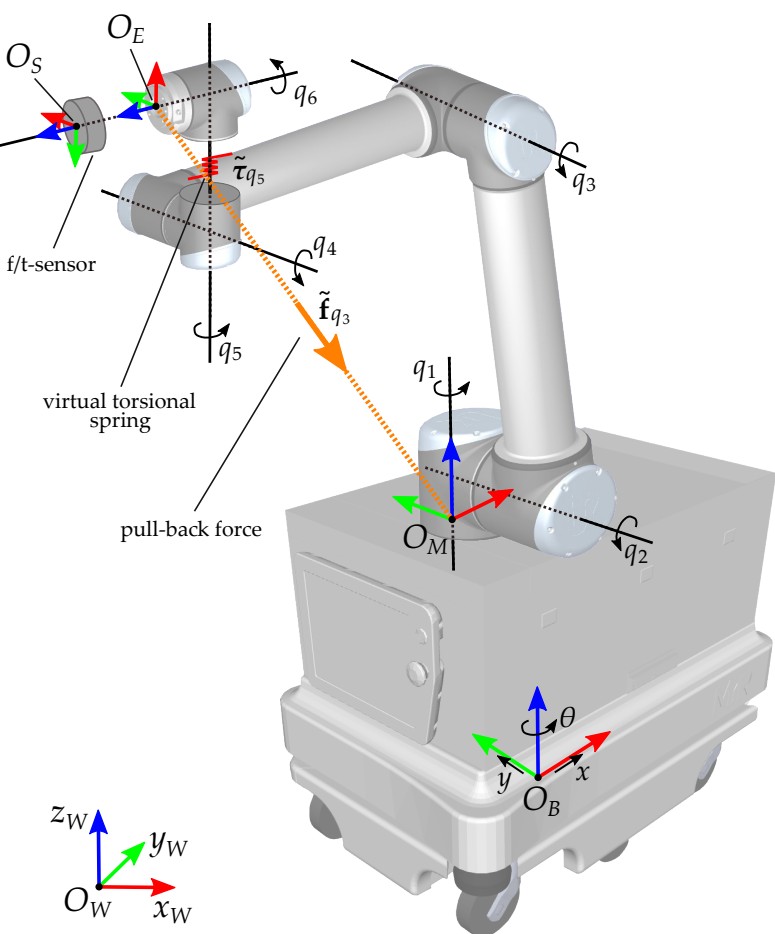

**Figure 2.** CHIMERA joints and coordinates: The mobile base is modelled with two linear joints $x$ and $y$ and one rotational joint $\Theta$. The UR-10 has six rotational joints denoted as $q_i$ with $i = 1, \dots, 6$. The Coordinate Systems are defined with their origins $O$ and three axis-vectors $x$, $y$ and $z$. Shown are the world-coordinate system $\Sigma_W := \{O_W; x_W, y_W, z_W\}$, the frame of the mobile base $\Sigma_B := \{O_B; x_B, y_B, z_B\}$, the UR-10 base frame $\Sigma_M := \{O_M; x_M, y_M, z_M\}$, the EE frame $\Sigma_E := \{O_E; x_E, y_E, z_E\}$ and the coordinate system of the force-torque sensor $\Sigma_S := \{O_S; x_S, y_S, z_S\}$. The virtual pull-back force for singularity avoidance in joint 3 is denoted as $\tilde{\mathbf{f}}_{q_3}$ and the virtual torque for singularity avoidance in joint 5 as $\tau_5$.

### 3.1. Distribution of Motion

The distribution of motion is realized as follows: Two circles, an inner and an outer one, are used to define three zones in the $xy$-plane of the robot base frame, as depicted in Figure 3. We switch between three main operation modes, depending on the position of the end effector (EE) in the $xy$-plane. If the EE is located between the two circles ($r_i < r < r_o$), only the serial manipulator moves, denoted as *UR-Mode*. Outside of the outer circle ($r > r_o$) we switch to *Pull-Mode*, where the mobile base can be pulled like a trailer and haptic user-feedback is realized by means of a virtual spring. This virtual spring generates a force to move the EE back inside the circle. When the EE enters the inner circle ($r < r_i$) we switch to *Push-Mode*. The user can move the base by pushing it and a virtual spring generates a force to move the EE back out of this inner circle.

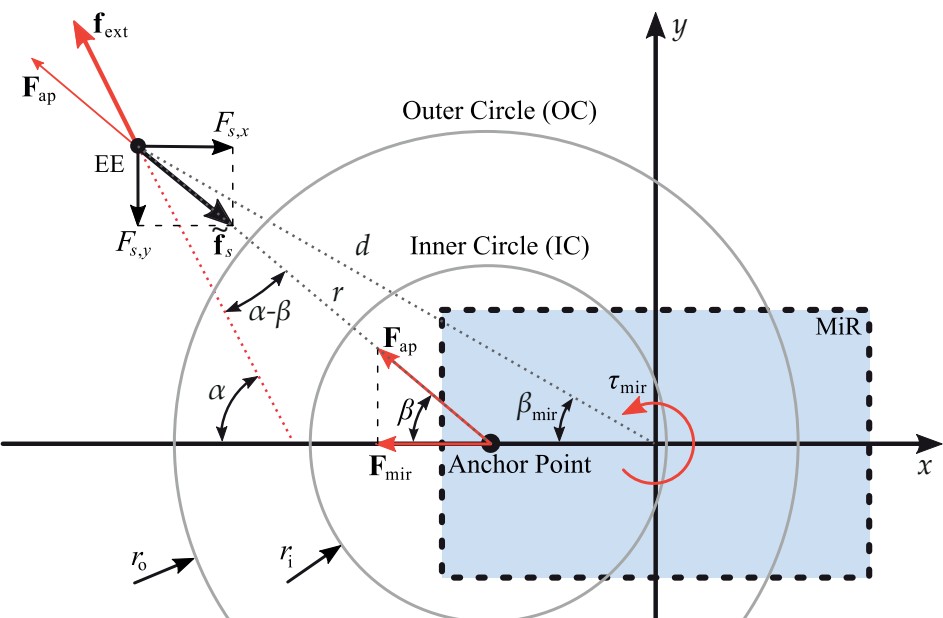

**Figure 3.** Kinetic relationships under external forces: This figure illustrates the angles and forces when the EE is outside the outer circle (*Pull-Mode*) and a force and torque is projected to the mobile base as described in Section 3.1.

The control inputs of the system are the EE-velocities $\dot{\mathbf{x}}_{ur}^{\Sigma_B} = \begin{bmatrix} \mathbf{v}^T & \boldsymbol{\omega}^T \end{bmatrix}^T \in \mathbb{R}^{6\times 1}$ and the velocities of a mobile base in the general case $\dot{\mathbf{q}}_{mir}^{\Sigma_B} = \begin{bmatrix} \dot{x} & \dot{y} & \dot{\Theta} \end{bmatrix}^T \in \mathbb{R}^{3\times 1}$, all given in the frame of the mobile base $\Sigma_B := \{O_B; x_B, y_B, z_B\}$. For simplicity, we drop the subscript for the reference coordinate, thus in the following, vectors without an explicit subscript are all given in the mobile-base-frame. In all modes, the controller equations are given by

$$\begin{bmatrix} \dot{\mathbf{x}}_{ur} \\ \dot{\mathbf{q}}_{mir} \end{bmatrix} = \begin{bmatrix} \mathbf{B}_{ur}^{-1} & \mathbf{0} \\ \mathbf{0} & \mathbf{B}_{mir}^{-1} \end{bmatrix} \begin{bmatrix} \mathbf{w}_{ext} + \tilde{\mathbf{w}}_{fb} \\ \tilde{\mathbf{w}}_{mir} \end{bmatrix} - \begin{bmatrix} \dot{\mathbf{x}}_c \\ \mathbf{0} \end{bmatrix} \tag{6}$$

where $\mathbf{B}_{ur} \in \mathbb{R}^{6\times 6}$ and $\mathbf{B}_{mir} \in \mathbb{R}^{3\times 3}$ are the diagonal positive definite damping matrices, $\mathbf{w}_{ext} = \begin{bmatrix} \mathbf{f}_{ext}^T & \boldsymbol{\tau}_{ext}^T \end{bmatrix}^T \in \mathbb{R}^{6\times 1}$ is the wrench vector, including external forces and torques applied to the EE, $\tilde{\mathbf{w}}_{fb} \in \mathbb{R}^{6\times 1}$ is a wrench vectors for haptic feedback including the virtual spring forces and singularity avoidance wrenches as described in Section 3.2 and $\tilde{\mathbf{w}}_{mir} = \begin{bmatrix} F_{mir} & 0 & \tau_{mir} \end{bmatrix}^T \in \mathbb{R}^{3\times 1}$ includes the projected force for linear motion and projected torque for angular motion of the mobile base as shown in Equation (7). The vector of EE-velocities to compensate for angular motions of the base is denoted as $\dot{\mathbf{x}}_c$. We assume that the applied wrench $\mathbf{w}_{ext}$ acting on the EE is known, either by

using a force-torque sensor or joint torque estimation based on motor current measurements (see, e.g., [22,23]).

Mode-dependent variables are the projected wrench $\tilde{\mathbf{w}}_{\mathrm{mir}}$ of the mobile base and the haptic feedback wrench $\tilde{\mathbf{w}}_{\mathrm{fb}}$. To move the mobile base, we project the applied external wrench to a linear pulling or pushing force $F_{\mathrm{mir}}$ and a rotation torque $\tau_{\mathrm{mir}}$. These projected values are only computed if the EE is not located in between the inner and the outer circle, e.g., in *Pull-Mode* and *Push-Mode*. Since the mobile base is non-holonomic due its the differential drive, no linear motion in $y$-direction is possible and the second entry of the projected wrench $\tilde{\mathbf{w}}_{\mathrm{mir}} = \begin{bmatrix} F_{\mathrm{mir}} & 0 & \tau_{\mathrm{mir}} \end{bmatrix}^{\mathrm{T}} \in \mathbb{R}^{3\times 1}$ is set to zero. This strategy is inspired by the design of a steered trailer, which most persons are familiar with. The projections are given as

$$
\begin{bmatrix} F_{\mathrm{mir}} \\ \tau_{\mathrm{mir}} \end{bmatrix} = \begin{cases} \begin{bmatrix} |\mathbf{f}_{\mathrm{ext}}|\cos(\alpha - \beta)\sin(\beta) \\ p_x|\mathbf{f}_{\mathrm{ext}}|\cos(\alpha - \beta)\cos(\beta) \end{bmatrix} & r > r_{\mathrm{out}} \text{ and } |\beta| - |\alpha| < \frac{\pi}{2} \quad (\textit{Pull-Mode}) \\[12pt] \begin{bmatrix} |\mathbf{f}_{\mathrm{ext}}|\cos(\alpha - \beta)\sin(\beta) \\ p_x|\mathbf{f}_{\mathrm{ext}}|\cos(\alpha - \beta)\cos(\beta) \end{bmatrix} & r < r_{\mathrm{in}} \text{ and } |\beta| - |\alpha| > \frac{\pi}{2} \quad (\textit{Push-Mode}) \\[12pt] \begin{bmatrix} 0 \\ 0 \end{bmatrix} & \text{otherwise} \quad (\textit{UR-Mode}) \end{cases}
\tag{7}
$$

with $p_x$ denoting the $x$-coordinate of the anchor point and the angles $\alpha$ and $\beta$ as illustrated in Figure 3. The additional conditions that consider the angles $\alpha$ and $\beta$ in Equation (7) ensure that only forces in the desired direction, based on the actual mode, are projected to the base (e.g., no pushing of the base in *Pull-Mode*). The projected force and torque are then transferred to motion as described in Equation (6). The translational motion of the EE in world coordinates that is caused by a translational motion of the base feels natural and as intended when interacting with the robot. In contrast, rotations of the base cause the hand guided EE to push towards a side, which feels unexpected and unnatural, thus this motion must be compensated. The compensation vector is given by $\mathbf{v}_{\mathrm{c}} = \begin{bmatrix} v_{\mathrm{c,x}} & v_{\mathrm{c,y}} & \mathbf{0}^{\mathrm{T}} \end{bmatrix}^{\mathrm{T}}$ with $v_{\mathrm{c,x}}$ and $v_{\mathrm{c,y}}$ as the linear velocities of the EE in $x$ and $y$ direction and $\mathbf{0}^{\mathrm{T}}$ a $4\times 1$ zero vector. The components can be determined as

$$
\begin{bmatrix} v_{\mathrm{c,x}} \\ v_{\mathrm{c,y}} \end{bmatrix} = \begin{bmatrix} -d\,\dot{\theta}\sin(\beta_{\mathrm{mir}}) \\ d\,\dot{\theta}\cos(\beta_{\mathrm{mir}}) \end{bmatrix}.
\tag{8}
$$

### 3.2. Haptic Feedback

The haptic feedback provided to the user fulfills several purposes. First, whenever the EE leaves the space between the two circles, so *Push-* or *Pull-Mode* is active, a virtual spring force is generated. This provides the naturally expected resistance when pulling or pushing the mobile base. Second, to avoid approaching singular arm configurations. The avoidance of the shoulder singularity is already guaranteed by means of the inner circle. The remaining two causes for a singularity, a fully stretched elbow (joint 3) and a critical wrist configuration (joint 5), are avoided by adding additional virtual feedback wrenches whenever one of these joint-position gets too close to a critical value. The total wrench-vector for haptic feedback

$$
\tilde{\mathbf{w}}_{\mathrm{fb}} = \tilde{\mathbf{w}}_{\mathrm{s}} + \tilde{\mathbf{w}}_{\mathrm{q}_3} + \tilde{\mathbf{w}}_{\mathrm{q}_5}
\tag{9}
$$

is determined as the sum of the wrench $\tilde{\mathbf{w}}_{\mathrm{s}}$ including the virtual spring forces in *Pull-* or *Push-Mode* and $\tilde{\mathbf{w}}_{\mathrm{q}_3}$ and $\tilde{\mathbf{w}}_{\mathrm{q}_5}$ for singularity-avoidance in joints 3 and 5, respectively.

### 3.2.1. Virtual Spring

The borders between the three different zones are defined as circles in the $xy$-plane as shown in Figure 3, resulting in cylindrical shapes in 3D-space, since the $z$-coordinate of the EE is not taken into account here. Thus, also the virtual spring force acts in the $xy$-plane only, consequently $\tilde{\mathbf{w}}_s = \begin{bmatrix} F_{s,x} & F_{s,y} & \mathbf{0}^T \end{bmatrix}^T$, where $F_{s,x}$ and $F_{s,y}$ are the $x$ and $y$ components, respectively, and $\mathbf{0}$ denotes the $4 \times 1$ zero vector. The equations to determine these components are given by

$$
\begin{bmatrix} F_{s,x} \\ F_{s,y} \end{bmatrix} = \begin{cases} \begin{bmatrix} -k_{\text{pull}} \cos(\beta)(r - r_{\text{o}}) \\ -k_{\text{pull}} \sin(\beta)(r - r_{\text{o}}) \end{bmatrix} & r > r_{\text{o}} \ \ (\textit{Pull-Mode}) \\[2ex] \begin{bmatrix} -k_{\text{push}} \cos(\beta)(r - r_{\text{i}}) \\ -k_{\text{push}} \sin(\beta)(r - r_{\text{i}}) \end{bmatrix} & r < r_{\text{i}} \ \ (\textit{Push-Mode}) \\[2ex] \begin{bmatrix} 0 \\ 0 \end{bmatrix} & \text{otherwise} \ \ (\textit{UR-Mode}) \end{cases} \tag{10}
$$

with $k_{\text{pull}}$ and $k_{\text{push}}$ as the spring constants of the virtual springs, $r_o$ and $r_i$ as the radii of the inner and outer circles, respectively, $r$ as the $xy$-distance between $O_E$ and $O_M$ and the angle $\beta$ as depicted in Figure 3.

### 3.2.2. Singularity Avoidance

As discussed in Section 2 there are three types of singularities: The shoulder singularity, the elbow singularity and the wrist singularity. The shoulder singularity is already avoided with the inner circle. Whenever the EE enters this inner circle, a force pointing in the opposite direction is generated, thus by choosing $r_i$ sufficiently large the point $P_{56}$ (see Section 2) cannot reach the plane spanned by the axis of the first and second joint in the base frame of the serial manipulator, despite applying immensely high forces which assume the user will not do.

With a fully stretched elbow, the EE looses its ability to move further away from its base and the arm is in a singular configuration. We avoid this by applying a force to the EE with direction back to origin of the base of the serial manipulator whenever the elbow (joint 3) get closer than a specified distance to the critical joint position, as depicted in Figure 4. The direction of the force is therefore given by the unit-vector $-\mathbf{e}_{O_E}$, which is the negative normalized translation vector of the EE in $\Sigma_M$. The pullback-force is determined as

$$
\tilde{\mathbf{f}}_{q_3} = \begin{cases} -\mathbf{e}_{O_E} k_3 (q_3 - t_3) & q_3 > t_3 \\ \mathbf{0} & \text{otherwise} \end{cases} \tag{11}
$$

and its magnitude increases, the more the elbow gets stretched. We do not want any feedback torques here, thus $\tilde{\boldsymbol{\tau}}_{q_3} = \mathbf{0}$. The wrench vector for haptic feedback to avoid the elbow singularity is then given by

$$
\tilde{\mathbf{w}}_{q_3} = \begin{bmatrix} \tilde{\mathbf{f}}_{q_3} \\ \mathbf{0} \end{bmatrix}. \tag{12}
$$

The wrist singularity occurs, whenever the second wrist joint (joint 5) approaches the position $k\pi$, $k \in \mathbb{Z}$, causing the rotation axes of the other two wrist joints (joints 4 and 6) being parallel. Similar to the avoidance technique for the elbow singularity, we specify a threshold for the minimum distance to the critical joint position. As shown in Figure 4, when the distance falls below this threshold, a virtual torque in the 5-th joint is generated by means of a torsional spring to prevent coming too close to the singular position. The virtual torque is determined as

$$\tau_5 = \begin{cases} k_5(q_5 - t_{5,\text{low}}) & q_5 < t_{5,\text{low}} \\ k_5(q_5 - t_{5,\text{hi}}) & q_5 > t_{5,\text{hi}} \\ 0 & \text{otherwise} \end{cases} \tag{13}$$

where $\tau_5$ is the torque caused by the virtual spring, $k_5$ denotes the stiffnesses of the virtual torsional spring, $q_5$ is the angular position of the joint and $t_{i,\text{hi}}$ and $t_{i,\text{low}}$ are the upper and lower thresholds for the virtual spring to become active.

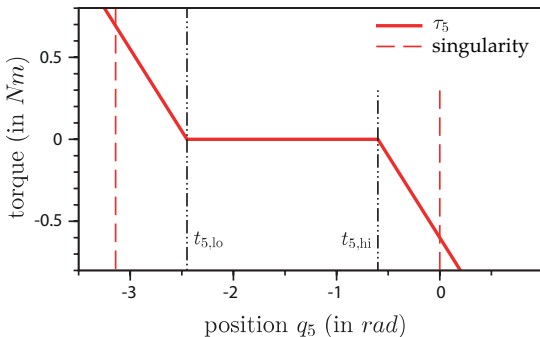 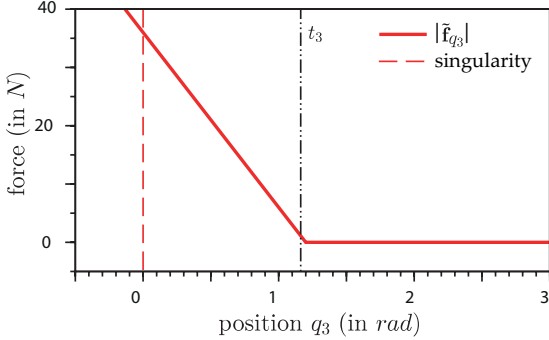

**Figure 4.** Virtual values for singularity-avoidance: Left: Virtual torque in joint 5. Right: Virtual pullback-force caused by the elbow joint (joint 3).

This virtual torque in the 5-th joint has to be transformed to an associated EE wrench. To determine the reactive force we need the Jacobian $\mathbf{J}$, which is a function of the joint positions $\mathbf{q}_{ur}$ and composed of a linear part $\mathbf{J}_v$ and a rotational part $\mathbf{J}_\omega$, consequently

$$\mathbf{J} = \begin{bmatrix} \mathbf{J}_v \\ \mathbf{J}_\omega \end{bmatrix} = \begin{bmatrix} \mathbf{j}_{v,q_1} & \mathbf{j}_{v,q_2} & \mathbf{j}_{v,q_3} & \mathbf{j}_{v,q_4} & \mathbf{j}_{v,q_5} & \mathbf{j}_{v,q_6} \\ \mathbf{j}_{\omega,q_1} & \mathbf{j}_{\omega,q_2} & \mathbf{j}_{\omega,q_3} & \mathbf{j}_{\omega,q_4} & \mathbf{j}_{\omega,q_5} & \mathbf{j}_{\omega,q_6} \end{bmatrix}. \tag{14}$$

With the Jacobian we can determine the EE-velocities for a given set of joint-speeds. In particular, we are interested in the linear EE-velocities caused by 5-th joint, which is given in $\mathbf{j}_{v,q_5}$. The reactive force at the EE, caused by a given torque around the axis of rotation of the 5-th joint is indirectly proportional to the distance $|\mathbf{j}_{v,q_5}|$, thus we need to invert the magnitude of this vector while maintaining the same direction. This resulting vector is also known as the Samelson inverse and the reactive force is determined as:

$$\tilde{\mathbf{f}}_{q_5} = \frac{\mathbf{j}_{v,q_5}}{|\mathbf{j}_{v,q_5}|^2}\tau_5 \tag{15}$$

To achieve the desired motion around this axis, the chosen damping coefficients of our controller

$$\mathbf{B}_{ur} = \begin{bmatrix} \mathbf{B}_v & \mathbf{0} \\ \mathbf{0} & \mathbf{B}_\omega \end{bmatrix} \tag{16}$$

need to be taken into account. As given in Equation (6), without external forces ($\mathbf{f}_{\text{ext}} = \mathbf{0}$), the linear velocity-vector $\mathbf{v}$ of the EE, as a reaction to the virtual force $\tilde{\mathbf{f}}_{q_5}$ is given by

$$\mathbf{v} = \mathbf{B}_v^{-1}\tilde{\mathbf{f}}_{q_5}. \tag{17}$$

To Keep the EE on the desired circular trajectory around the axis of rotation of joint 5, the relation between linear and angular velocities

$$\mathbf{v} = |\mathbf{j}_{v,q_5}|\boldsymbol{\omega} \tag{18}$$

must hold. The angular EE-velocities are determined by the controller as

$$\boldsymbol{\omega} = \mathbf{B}_\omega^{-1} \tilde{\boldsymbol{\tau}}_{\mathrm{q5}} \tag{19}$$

and thus, to satisfy the constraint from Equation (18), the feedback-torque at the EE is given with

$$\tilde{\boldsymbol{\tau}}_{\mathrm{q5}} = \frac{1}{|\mathbf{j}_{\mathrm{v,q5}}|} \mathbf{B}_\omega \mathbf{B}_\mathrm{v}^{-1} \tilde{\mathbf{f}}_{\mathrm{q5}}. \tag{20}$$

The wrench-vector for the haptic feedback of the virtual torsional spring in joint $q_5$ is given by

$$\tilde{\mathbf{w}}_{\mathrm{q5}} = \begin{bmatrix} \tilde{\mathbf{f}}_{q_5} \\ \tilde{\boldsymbol{\tau}}_{q_5} \end{bmatrix}. \tag{21}$$

## 4. Experimental Results

To show the effectiveness of the proposed control structure several laboratory experiments were carried out (see supplementary video). This includes straight pulling (Section 4.1) and pushing (Section 4.2) manoeuvres of the EE to demonstrate the working principal of the motion-distribution between serial manipulator and mobile base. A curved pulling experiment (Section 4.3) shows that the mobile manipulator behaves similarly to a simple steered trailer, which we used as inspiration for the controller design. We also show detailed results of the singularity avoidance techniques. As mentioned in Section 3.2.2, the shoulder singularity is avoided by means of the virtual spring of the inner circle. Even tough this is a restrictive choice and permits a large area of the workspace of the serial manipulator it prevents the arm from approaching the shoulder-singularity and no explicit experiments were performed for this case. Results for avoiding the elbow and wrist singularities are discussed in Sections 4.4 and 4.5, respectively. The threshold values $t_3, t_{5,\mathrm{lo}}, t_{5,\mathrm{hi}}$, the elements of the damping matrices $\mathbf{B}_v$, $\mathbf{B}_\omega$, $\mathbf{B}_\mathrm{mir}$ as well as the parameters $k_\mathrm{pull}$, $k_\mathrm{push}$, $k_3$, $k_5$ were determined empirically. All parameters used for the experiments are given in Table 1.

**Table 1.** Table of parameters

| Symbol | Value | Unit | Description |
|---|---|---|---|
| $r_\mathrm{i}$ | 0.48 | m | Radius of inner circle |
| $r_\mathrm{o}$ | 0.8 | m | Radius of outer circle |
| AP | $\begin{bmatrix} -0.28 & 0 & 0.6 \end{bmatrix}$ | m | Anchor-point in $\Sigma_B$ |
| $\mathbf{B}_\mathrm{v}$ | $\begin{bmatrix} 40 & 0 & 0 \\ 0 & 40 & 0 \\ 0 & 0 & 40 \end{bmatrix}$ | N·s/m | Translational damping matrix |
| $\mathbf{B}_\omega$ | $\begin{bmatrix} 2 & 0 & 0 \\ 0 & 2 & 0 \\ 0 & 0 & 2 \end{bmatrix}$ | Nm·s/rad | Rotational damping matrix |
| $\mathbf{B}_\mathrm{mir}$ | $\begin{bmatrix} 50 & 0 & 0 \\ 0 & 1 & 0 \\ 0 & 0 & 7 \end{bmatrix}$ | - | Mobile base damping matrix |
| $k_\mathrm{pull}$ | 140 | N/m | Virt. spring stiffness *Pull-Mode* |
| $k_\mathrm{push}$ | 300 | N/m | Virt. spring stiffness *Push-Mode* |
| $k_3$ | 30 | - | Constant for pushback-force |
| $k_5$ | 1 | N/rad | Virt. spring stiffness in joint 5 |
| $t_3$ | 1.2 | rad | Position threshold for joint 3 |
| $t_{5,\mathrm{lo}}$ | $-2.45$ | rad | Position threshold for joint 5 |
| $t_{5,\mathrm{hi}}$ | $-0.6$ | rad | Position threshold for joint 5 |
| $\mathbf{a}_\mathrm{dh}$ | $\begin{bmatrix} 0 & 0.6127 & 0.5716 & 0 & 0 & 0 \end{bmatrix}$ | m | DH-Parameters of UR-10: $a$ |
| $\mathbf{d}_\mathrm{dh}$ | $\begin{bmatrix} 0.118 & 0 & 0 & 0.163941 & 0.1157 & 0.0922 \end{bmatrix}$ | m | DH-Parameter of UR-10: $d$ |
| $\boldsymbol{\alpha}_\mathrm{dh}$ | $\begin{bmatrix} \pi/2 & 0 & 0 & \pi/2 & -\pi/2 & 0 \end{bmatrix}$ | rad | DH-Parameter of UR-10: $\alpha$ |

### 4.1. Straight Pulling

The results of straight pulling manoeuvre are shown in Figure 5. The EE starts between the two circles and the controller is in *UR-Mode*, thus the applied force $\mathbf{f}_{ext}$ at the EE initially only causes a motion of the EE. As the radius $r$ increases and the EE leaves the outer circle (first vertical green line), a switch to *Pull-Mode* arises and the applied forces are also projected to the mobile base and cause motion. Withing this experiment, the EE was tried to pull along the negative $x$-axis, thus the angle $\beta$ was very small (See Figure 3). As a result, the magnitudes of the projected torque $\tau_{mir}$ and the angular velocity $\dot\theta$ of the mobile base are small. Once no more force is applied and the EE is released (second vertical green line) the base stops and the EE moves back inside the outer circle.

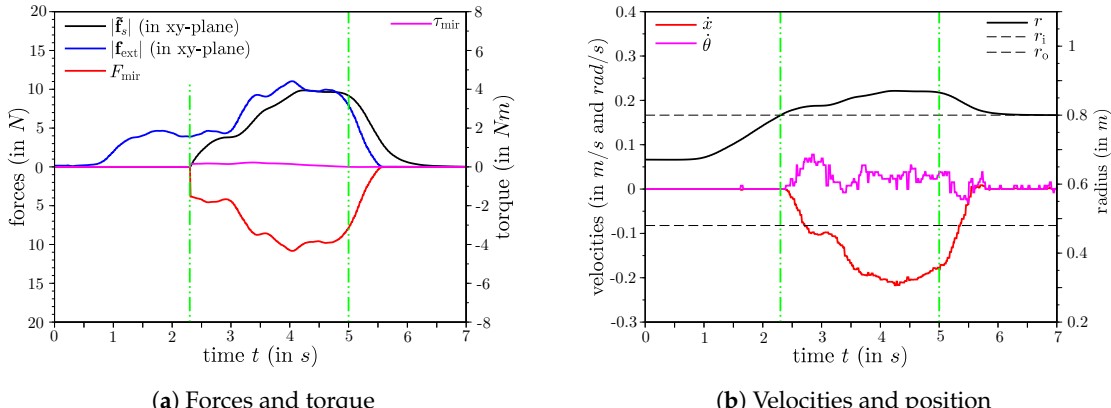

(**a**) Forces and torque            (**b**) Velocities and position

**Figure 5.** Results of a straight pulling manoeuvre: Plot (a): Left axis includes the norm of the external virtual spring torque vector $|\tilde{\mathbf{f}}_s|$ (black), the norm of the external force $|\mathbf{f}_{ext}|$ (blue) and the projected force for the mobile base $F_{mir}$ (red). Right axis shows the projected torque for the mobile base $\tau_{mir}$ (magenta). Plot (b): Left axis includes the linear velocity of the mobile base $\dot x$ (red) and its rotational velocity $\dot\theta$ (magenta). Right axis shows the radius $r$, which is the $xy$-distance between EE and UR10 base (black solid) and the radii $r_i$ and $r_o$ of the inner and outer circles (dashed black), respectively.

### 4.2. Straight Pushing

In Figure 6, the results of a straight pushing manoeuvre are shown. In this experiment, the EE is pushed along the $x$-axis towards the anchor point. Similar to the straight pulling experiment, the EE starts between the two circles and within the first few seconds only the robotic arm moves until the EE enters the inner circle (first vertical green line). The user receives haptic feedback by the means of the virtual spring with increasing magnitude the deeper the EE enters the inner circle. At the same time, a force and a torque are projected to the mobile base and causes motion there. We tried to push the EE along the x-axis, thus also here the magnitudes of the projected torque and angular velocity of the mobile base are relatively low compared to the curved pulling experiment. As Soon as the EE is released it returns to the inner circle and the mobile base stops.

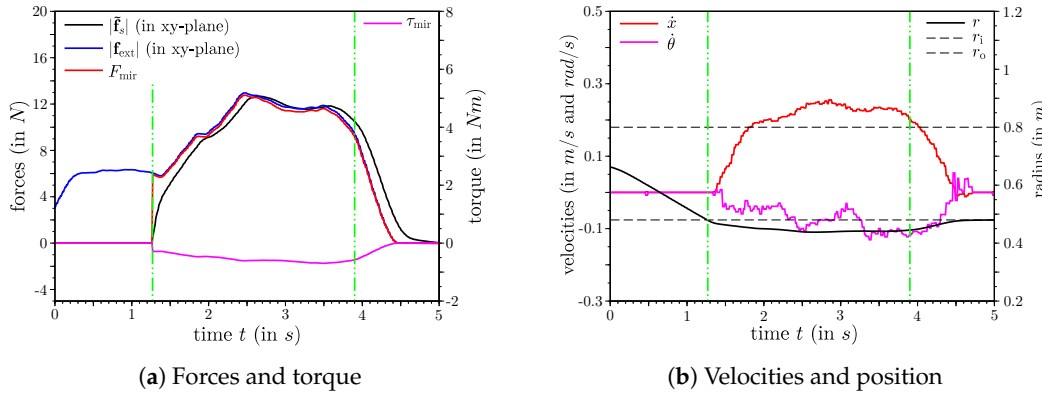

(**a**) Forces and torque  (**b**) Velocities and position

**Figure 6.** Results of a straight pushing manoeuvre.

### 4.3. Curved Pulling

For this experiment, a curved pulling action is performed with the results shown in Figure 7. In contrast to the last two experiments, where pulling or pushing happened along the *x*-axis ($\beta \approx 0$), the EE is pulled with an angle, so that a higher projected torque is generated once the EE leaves the outer circle. This torque causes an angular velocity of the base so that is turns towards the pulling direction. The amplitude of the rotational velocity decreases the closer the EE gets towards the negative x-axis again. Once the base faces the direction only the translational motion remains until releasing the EE.

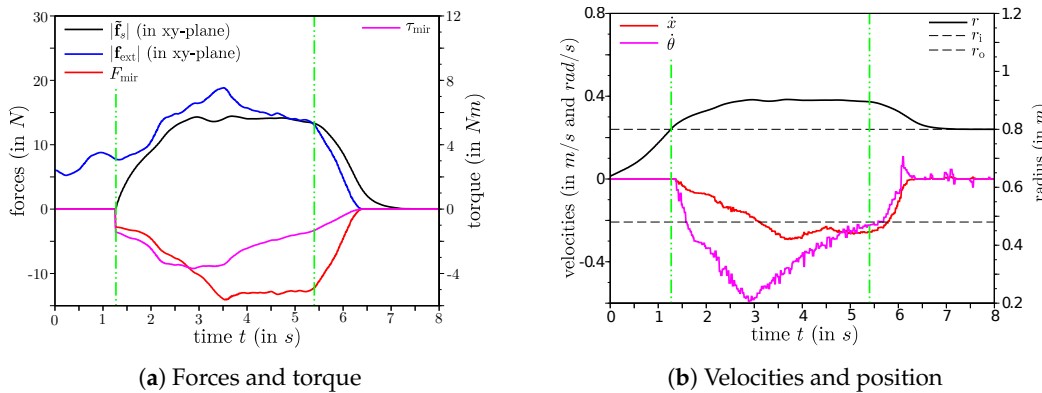

(**a**) Forces and torque  (**b**) Velocities and position

**Figure 7.** Results of a curved pulling manoeuvre.

### 4.4. Singularity Avoidance–Elbow-Joint

To show the effectiveness of the proposed technique to avoid a singular configuration caused by the elbow joint, two similar experiments were performed: One with active singularity-avoidance, shown in Figure 8a, and a second with inactive singularity-avoidance ($|\tilde{\mathbf{f}}_{q_3}| = 0$), shown in Figure 8b. For this experiment, the EE starts between the two circles (*UR-mode*) and is pulled upwards. The inner and outer circles are defined in the *xy*-plane, which results in cylindrical borders in the 3D-space. Without singularity-avoidance it is possible to move the EE in between these cylindrical borders freely, so there is no limitation on the height. This could result in a fully stretched elbow causing a singular arm configuration as demonstrated in Figure 8b (second green line). Please note that within this second experiment no pullback-force is applied when $q_3$ falls below the threshold value $t_3$. As a result to the applied pulling-force the elbow stretches more and more until it hits the critical position and the UR10-controller goes into protective stop. The results also show an increasing joint velocity $\dot{q}_3$ as $q_3$ gets closer to the critical position. This fast joint movement could be very dangerous for humans near the robot and must be avoided. With active singularity-avoidance a pullback-force is applied after the threshold is hit (first green line in Figure 8a) preventing $q_3$ getting close to the critical position.

During our experiments, it was not possible to get a fully stretched elbow even when excessively high pulling-forces were applied by the user. The working principal of this technique is also depicted in Figure 9.

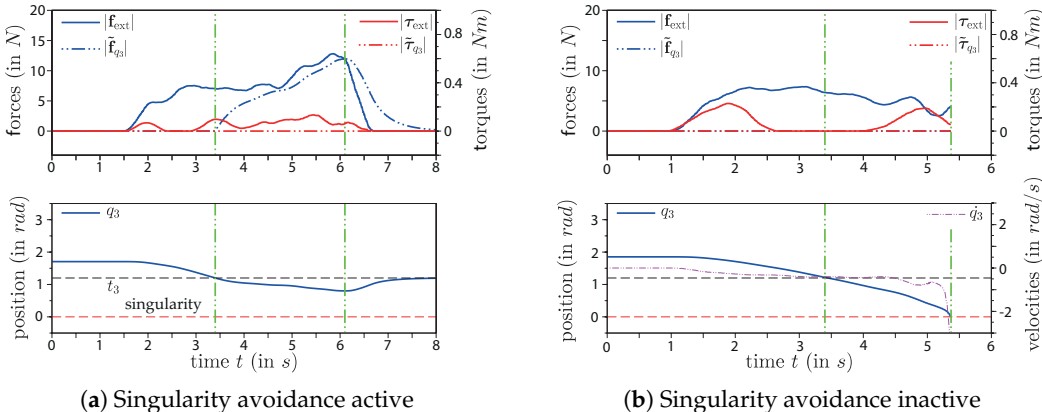

(**a**) Singularity avoidance active  (**b**) Singularity avoidance inactive

**Figure 8.** Elbow-singularity avoidance. Upper plots: Left axis include the norms of the external force $|\mathbf{f}_{\text{ext}}|$ (blue) and of the virtual pullback-force $|\tilde{\mathbf{f}}_{q_3}|$ (blue dash-dotted). Right axis include the norms of the external torque $|\boldsymbol{\tau}_{\text{ext}}|$ (red) and of the virtual torque $|\tilde{\boldsymbol{\tau}}_{q_3}|$ (red dash-dotted). The lower plots show the joint position $q_3$ (blue), the position-threshold $t_3$ (black dashed) and the singular position (red dashed) on the left axis. The lower right plot also shows the joint velocity $\dot{q}_3$ (magenta dash-dotted) on the right axis.

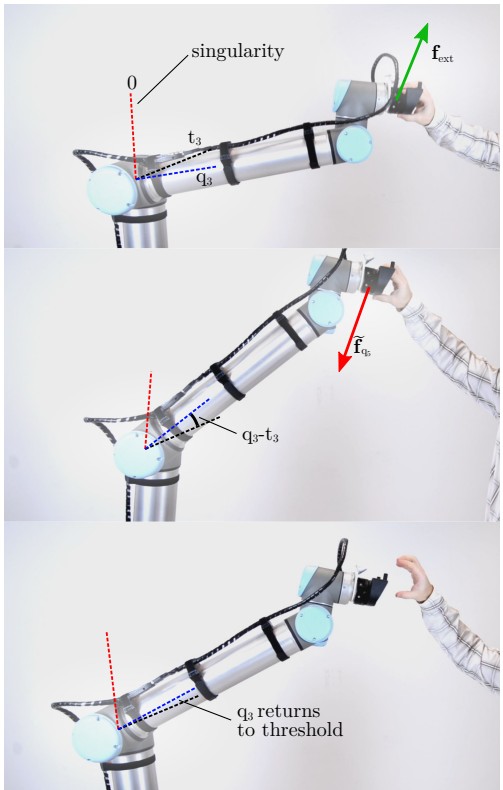

**Figure 9.** Singularity avoidance in the 3rd joint. The EE is pulled away from the base (top image) causing the elbow joint to move towards the stretched position. As soon as the joint position surpasses the specified threshold $t_3$ a virtual force is applied to the EE pointing back to the base (middle image). This force increases the more the elbow stretches. Thus, without applying extremely high forces it is not possible for user to get into the singular position of the 3rd joint. After releasing the EE, is moves back towards the base until the position $q_3$ reaches the threshold (bottom image).

*4.5. Singularity Avoidance - Wrist-Joint*

The results of the proposed singularity-avoidance strategy for the wrist joint (joint 5) are shown in Figure 10a. For comparison, a similar experiment with inactive singularity-avoidance was carried out with the results shown in Figure 10b. During this experiments, the robot started in a configuration with the joint position $q_5$ near the upper threshold $t_{q_5,\text{hi}}$ and the EE was pushed back towards the base, as depicted in Figure 11. With active singularity avoidance, a virtual force $\tilde{\mathbf{f}}_{q_5}$ and a virtual torque $\tilde{\boldsymbol{\tau}}_{q_5}$ are applied to the EE after $q_5$ surpasses the threshold (first vertical green line in Figure 10a). As the upper plot shows, the applied and virtual torques have almost the same magnitude. For better readability, only the norms of these vectors are plotted, but these torques are around the same axis but in opposite direction. Thus, they cancel each other in terms of EE motion generation in our controller equation given in Equation (6). Consequently the 5th joint is prevented from rotation further towards the critical position. When the EE is released joint 5 moves back to the threshold. When the singularity avoidance is turned off, the same manoeuvre results in further rotation of the 5th joint towards the singular position, as shown in Figure 10b. This plot also shows that the joint velocities drastically increase when $q_5$ gets near the critical position (second vertical green line in Figure 10b) which should be avoided in any case.

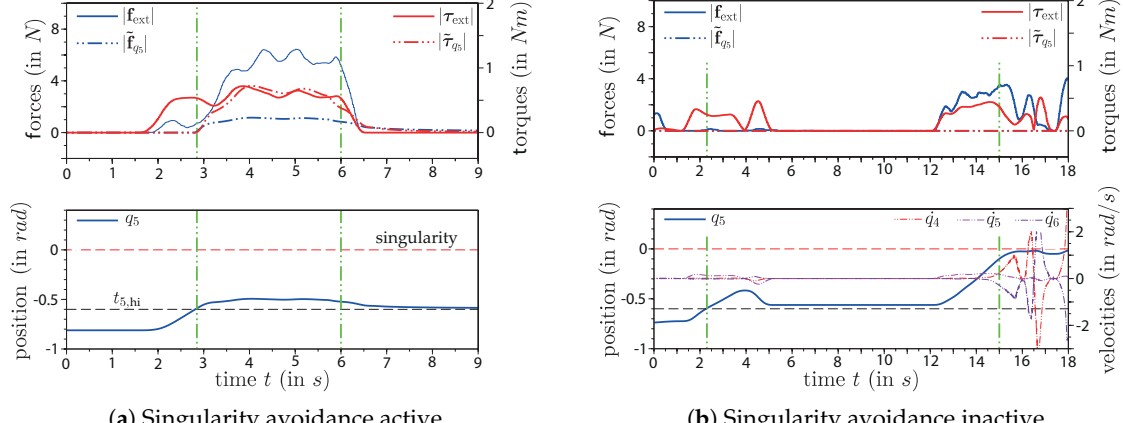

(**a**) Singularity avoidance active	(**b**) Singularity avoidance inactive

**Figure 10.** Wrist-singularity avoidance. Upper plots: Left axis include the norms of the external force $|\mathbf{f}_{\text{ext}}|$ (blue) and of the virtual force $|\tilde{\mathbf{f}}_{q_5}|$ (blue dash-dotted). Right axis include the norm of the external torque $|\boldsymbol{\tau}_{\text{ext}}|$ (red) and of the virtual torque $|\tilde{\boldsymbol{\tau}}_{q_5}|$ (red dash-dotted). The lower plots show the joint position $q_5$ (blue), the position-threshold $t_{5,\text{hi}}$ (black dashed) and the singular position (red dashed) on the left axis. The lower right plot also shows the joint velocities $\dot{q}_4$, $\dot{q}_5$ and $\dot{q}_6$ (red, magenta and blue dash-dotted) on the right axis.

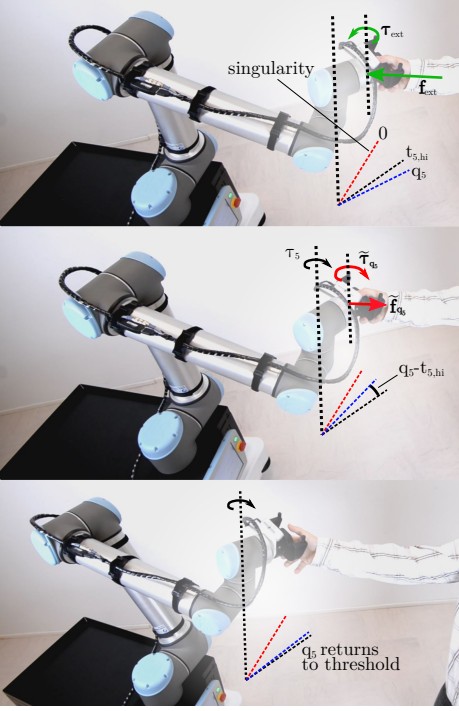

**Figure 11.** Singularity avoidance in the 5th joint. During this manoeuvre, the EE is pushed towards the mobile base. The User applies the external force $\mathbf{f}_{ext}$ and torque $\boldsymbol{\tau}_{ext}$. This causes a linear and angular motion of the EE. As soon as $q_5$ surpasses the threshold, a virtual torque and the corresponding virtual wrench at the EE are computed (middle image). As also described in Section 4.5, the external torque and the virtual torque for singularity avoidance cancel each other out and there is no more rotational motion. A translational motion towards the center remains, but it is not possible for user to get into the singular position of the 5th joint.

## 5. Conclusions

In this work, the practical use of a mobile manipulator was studied and demonstrated. We gave a detailed analysis of all possible singularities for the whole UR robot family and specifically pointed out those of the UR10. We proposed a control structure for hand-guiding the EE in Cartesian coordinates while handling both, the kinematic redundancies of the mobile manipulator and singular configurations of the robot arm. The conducted laboratory experiments on our mobile manipulator CHIMERA show that the system robustly permits these critical arm configuration while allowing the user to guide the EE to the desired target. It is also possible to either move the whole mobile manipulator or only the arm with fixed position of the mobile base without the need for any buttons or additional user interfaces. Moreover, the haptic feedback provided to the user by means of virtual forces and torques makes the interaction very intuitive and easy also for inexperienced users. This system design enables intuitive programming of mobile manipulator tasks using the Programming by Demonstration technique. Additionally the robot can be used as an assistant system without limitations on the workspace, e.g., for gravity compensation tasks. While investigations of the elbow and wrist singularities are straight forward, because each of them solely depends on one particular joint position, analyzing the shoulder singularity is more complex. We showed that our system avoids this configuration, but in a restrictive way since we deny a relatively large area of the manipulator's workspace.

For future work, we plan to refine the avoidance strategy especially for the shoulder singularity. By specifying a metric for the distance to the singularity the volume of the denied workspace could be reduced. Moreover, there are multiple solutions for the inverse kinematics of the serial manipulator. Switching from one posture the another implies going through a singularity and the current system design does not allow for manually switching the configuration (e.g., from elbow-up to elbow-down).

A singularity transition strategy could therefore also be useful to overcome this issue. Furthermore, we plan to eliminate the force-torque sensor on the EE. This means that we use the estimated external wrench based on the joint sensor values instead.

**Supplementary Materials:** The following are available online at www.mdpi.com/2218-6581/8/1/14/s1, a Video of the conducted experiments is included.

**Author Contributions:** M.W. and M.B. conceived and designed the control structure. M.H. performed the singularity analysis. M.W. performed the implementation on the robot and the experiments. M.W. and M.B. analyzed the data. All authors contributed to the writing process.

**Funding:** This research was funded by the Austrian Ministry for Transport, Innovation and Technology (BMVIT) within the framework of the sponsorship agreement formed for 2015-2018 under the project RedRobCo.

**Conflicts of Interest:** The authors declare no conflict of interest.

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
