# Peer review of "Singularity Avoidance Control of a Non-Holonomic Mobile Manipulator for Intuitive Hand Guidanceâ€"

_robotics, doi:10.3390/robotics8010014_

Round 1
Reviewer 1 Report
This paper develops a technique for hand guiding of a mobile manipulator in task space by using a force-torque sensor, which is mount in the end-effector. The main contribution of this work is the detail analysis of robot singularity and the singularity avoidance of robot joints during the hand guiding. The paper is well written and its effective experimental results is clear shown.
Minor comments:
1. The differential drive mobile base has only 2 independent variables. Why q_mir be R^3 but not R^2 ?
2. A common approach for redundant robot singularity avoidance is adding regulation term (or damped least-square), what is the strength of the proposed approach?
3. The experimental results focus mainly on straight pulling and pushing motion. Based on the proposed control structure, what useful and practical robot-human interaction tasks could be applied?
4. There is a typing error in matrix G_i of Eq. (1), d_1 should be replaced with d_i.
Author Response
Please see the uploaded Word document.

Reviewer 2 Report
The paper entitled Singularity Avoidance Control of a non-Holonomic Mobile Manipulator for Intuitive Hand Guidance presents a novel approach consisting of a force sensor (placed at the end-effector) and a novel control structure for singularity avoidance by means of haptic feedback. The paper is clear, original, well-structured, well-written and well-referenced. There are some minor comments that will improve the final version of the paper:
1) Sentence 49 states that the authors of [15] use a self-defined distance-to-singularity. This is not completely true. The authors define, using the singular values of the Jacobian matrix, a measure of closeness to a singularity (not a distance) in the line of all works using the DLS strategy for the inverse kinematics problem (see, for instance, the work of Chiaverini et al. (1991)).
2) In section 2.2, when introducing the matrices M_i and G_i, a comment about the DH parameters is needed. They can be introduced as the design parameters or DH parameters (together with a small description of them) of the UR10 Robot. In fact, in the rest of the section, the terms design parameters and DH parameters are used indistinctly. Thus, this clarification is needed.
3) When the line transform matrix \overline{T}, it is multiplied by a vector $p$ that should be removed from the equation (seems to be a typo).
4) The first paragraph of section 3.1 shows a very interesting idea for distributing the redundancy along the entire mobile manipulator. However, when reading it, the reviewer has the feeling of having read a similar approach before but I did not find a particular reference. Is this idea completely originated from the authors or is a novel strategy based on previous works? In the latter case, please include some references.
5) A comment on how to select the thresholds for the minimum distance to the singularities and the gains $k_pull$ and $k_push$ should be added, either in sections 3.2.1 and 3.2.2 or in section 4 when describing the experimental result
Author Response
please see the uploaded Word document.
